# The Methylene Spacer Matters: The Structural and Luminescent Effects of Positional Isomerism of n-Methylpyridyltriazole Carboxylate Semi-Rigid Ligands in the Structure of Zn(II) Based Coordination Polymers

**DOI:** 10.3390/polym15040888

**Published:** 2023-02-10

**Authors:** Pilar Narea, Benjamín Hernández, Jonathan Cisterna, Alejandro Cárdenas, Pilar Amo-Ochoa, Félix Zamora, Gerzon E. Delgado, Jaime Llanos, Iván Brito

**Affiliations:** 1Departamento de Química, Facultad de Ciencias Básicas, Universidad de Antofagasta, Casilla 170, Antofagasta 1240000, Chile; 2Departamento de Química, Facultad de Ciencias, Universidad de Católica del Norte, Sede Casa Central, Av. Angamos 0610, Antofagasta 1240000, Chile; 3Departamento de Física, Facultad de Ciencias Básicas, Universidad de Antofagasta, Casilla 170, Antofagasta 1240000, Chile; 4Departamento de Química Inorgánica, Universidad Autónoma de Madrid, 28049 Madrid, Spain; 5Institute for Advanced Research Chemistry (IAdChem), Universidad Autónoma de Madrid, 28049 Madrid, Spain; 6Condensed Matter Physics Center (IFIMAC), Universidad Autónoma de Madrid, 28049 Madrid, Spain; 7Laboratorio de Cristalografía, Departamento de Química, Facultad de Ciencias, Universidad de Los Andes, Mérida 5101, Venezuela

**Keywords:** metal-organic framework, luminescence, coordination polymers

## Abstract

Two Zn(II) coordination polymers (CPs) based on n-methylpyridyltriazole carboxylate semi-rigid organic ligands (n-MPTC), with *n* = **3** (**L1**) and **4** (**L2**), have been prepared at the water n-butanol interphase by reacting Zn(NO_3_)_2_·4H_2_O with Na**L1** and Na**L2**. This allows us to systematically investigate the influence of the isomeric positional effect on their structures. The organic ligands were obtained by saponification from their respective ester precursors ethyl-5-methyl-1-(pyridin-3-ylmethyl)-1H-1,2,3-triazole-4-carboxylate (**P1**) and ethyl-5-methyl-1-(pyridin-4-ylmethyl)-1H-1,2,3-triazole-4-carboxylate (**P2**), resulting in their corresponding sodium salt forms, 3-MPTC, and 4-MPTC. The structure of the Zn(II) CPs determined by single-crystal X-ray diffraction reveals that both CPs have 2D supramolecular hydrogen bond networks. The 2D supramolecular network of [Zn(L1)]n (**1**) is built up by hydrogen bond interactions between oxygen and hydrogen atoms between neighboring n-methylpyridyltriazole molecules, whereas in [Zn(L2)·4H_2_O]n (**2**) the water molecules link 1D polymeric chains forming a 2D supramolecular aggregate. The structures of **1** and **2** clearly show that the isomeric effect in the semi-rigid ligands plays a vital role in constructing the Zn(II) coordination polymers, helped by the presence of the methylene spacer group, in the final structural conformation. The structures of **1** and **2** significantly affect their luminescent properties. Thus, while **2** shows strong emission at room temperature centered at 367 nm, the emission of **1** is quenched substantially.

## 1. Introduction

Coordination polymers (CPs) are a family of compounds formed by the integration of metal entities and organic ligands that have been widely studied in recent decades due to their interesting structural features, and their chemical and physical properties [1,2,3,4,5,6], such as gas adsorbents [7,8,9,10], ion exchange [11,12], luminescence [13,14,15], magnetic materials [16,17], or catalysis [18], among others, have been described. Their structural design and/or physicochemical properties largely depend on the selection and combination of the building blocks, i.e., the organic ligands (spacers) and metal ions (nodes) [19]. Considering that our compounds are two-dimensional, we will refer to them as coordination polymers. The properties of CPs are not only affected by their intrinsic structural coordination bonds, but also by supramolecular interactions that can be formed between different structural moieties. Therefore, the generation of CPs based on the use of new organic ligands with specific structural and/or physical features opens new alternatives for material designs with desired properties [20,21,22,23,24,25,26]. On the other hand, solid-state properties for these crystalline materials, such as thermal stability, and fluorescent emission, can be related to the structural conformation and the coordination modes of the metal center used to synthesize these compounds.

This work focuses on the controlled design of Zn(II) CPs by designing flexible organic ligands that act as building blocks favoring interesting topologies [27]. The fine selection of positionally isomeric organic ligands, and their role in the structure of the CP, is a relevant aspect to study since it can also affect their physical properties. For instance, the isomeric effect [28,29,30] over coordination metal complexes can generate the difference between molecular or extended solids [25,30,31] and can affect the luminescent and thermal properties. Specifically, the luminescent properties could be modified on demand by appropriately selecting these building blocks. Additionally, great attention is focused on the study of luminescent Zn(II) complexes in the solid state and in solutions. Luminescent Zn(II) complexes are used for the production of organic light-emitting diodes (OLED) [32,33,34,35,36,37]. Our previous studies show how the isomeric position of ethyl 5-methyl-1-(pyridin-3-yl)-1H-1,2,3-triazole-3-carboxylate/-4-carboxylate type organic ligands has allowed for the obtaining of monodimensional Cd(II) CPs to supramolecular 3D networks with the consequent modification of their emission [28,38,39,40,41]. We have now selected Zn(II) as the *d*^10^ metal center and, as the flexible organic ligand, the tecton, 5-methyl-1-(pyridine-3-ylmethyl)-1*H*-1,2,3-triazole -4-carboxylate (Na**L1**), and 5-methyl-1-(pyridin-4-ylmethyl)-1*H*-1,2,3-triazole -4-carboxylate (Na**L2**) two positional isomers. These two ligands have a non-rigid methyl spacer group (–CH_2_–), which plays an important role in the construction of the Zn(II) CPs, with an influence on its luminescent properties. Additionally, it is worth mentioning that the preparation of materials based on Zn(II) brings additional advantages based on non-toxic features and natural availability. The study includes the synthesis, structural characterization, and photophysical properties of the novel CPs and their comparison with the previous related ones based on Cd(II).

## 2. Results and Discussion

### 2.1. Syntheses

The syntheses of the precursors, **P1** and **P2**, were carried out according to the literature [42,43,44] using [3 + 2] dipolar cyclo addition between *n*-methylpyridyl azide (*n* = 3 or 4) and 1,3-dicarbonyl compounds, such as ethyl acetoacetate (Figure 1). In general, the ester precursor yields are moderate (35–70%); meanwhile, the sodium carboxylate salts, Na**L1** and Na**L2**, gave quantitative yields. The resulting compounds are hygroscopic and thus must be stored in a dry box or inert gas atmosphere. These compounds are very soluble in water and soluble in other protic solvents, such as ethanol or hot methanol.

On the other hand, the CPs with general formulae, [Zn(**L1**)]*_n_* (**1**) and [Zn(**L2**)·4H_2_O]*_n_* (**2**), were obtained following a similar process to that which has been previously described in the literature [45] (The slow diffusion of the solution of the reactants, Na**L** and Zn(NO_3_)_2_·4H_2_O, in immiscible solvents (*n*-BuOH/H_2_O), resulted in the formation at the interphase of single crystals of **1** and **2** suitable for SC-XRD analysis (Figure 2)). In general, both compounds were obtained with a moderate yield (30 and 40%, respectively). Moreover, both compounds are stable to moisture and common atmospheric conditions and exhibit good thermal stability until 350 °C (Figure 8). Both CPs are insoluble in common organic solvents and water at room temperature. 

### 2.2. Crystallographic Studies

The powder X-ray patterns of **1** and **2** were indexed in monoclinic cells, which were corroborated by the X-ray single-crystal results. For each compound, a Rietveld refinement [46] was carried out using the Fullprof program [47] to check the structural parameters. Appendix A shows the perfect fit between the observed and calculated patterns. These results verify the phase purity of the crystalline samples and that the single crystals studied are representative of the bulk samples.

The single-crystal X-ray diffraction method established the crystal structures and chemical compositions of all compounds. The molecular structures of **L1** and **L2** are shown in their ester forms **P1** and **P2** (Figure 1). **P1** crystallizes in the monoclinic system with space group P2_1_/c, and **P2** crystallizes in an orthorhombic system with space group P2_1_2_1_2_1_, with both compounds having four molecular entities *per* unit cell with centrosymmetric and non-centrosymmetric settings, respectively. All the bond lengths and angles fall in the expected range for related compounds [48]. The C1–C6–N2 torsion angle for **P1** and **P2**, 112.80(19) and 113.1(2)°, respectively, shows a loss of coplanarity from the related heterocycles (*n*-pyridyl and 1,2,3-triazole moieties), 66.72(11) and 97.99(11)°, respectively. This is because the presence of the CH_2_ group allows a significant torsion angle between their heterocycles concerning similar compounds previously reported with the same moieties [49]. The main difference between both compounds is the orientation of the pyridyl fragment relative to the triazole-carboxylate moiety. The torsion angles C2/C1/C6/N2 are 75.7(2) and 22.2(4)° for **P1** and **P2**, respectively. Another significant difference between them is the position of the carbonyl group relative to a methyl group, *cis* in **P1**, and *trans* in **P2**, which in **P1** is stabilized by a C–H‧·‧O intramolecular hydrogen bond interaction with graph-set notation S(6) [50].

Figure 2 shows intermolecular C(4)–H(4)···O(1)^i^ hydrogen bond interactions that give rise to centrosymmetric dimers with a graph-set notation [50] along the crystal structure of **P1**. 

In the crystal structure of **P2** (Figure 3), the molecules are linked by C11–H11C⋅⋅⋅N1^i^ hydrogen bonds forming a *zig-zag* chain along the [001] direction with graph-set notation C(13). Further, these chains are linked to neighbor chains through C(12)–H(12A)⋅⋅⋅N(3), C(6)–H(6B)⋅⋅⋅N(4), and C(6)–H(6B)⋅⋅⋅O(1) hydrogen bonds, forming a supramolecular 2D ladder along the [010] direction.

The molecular structure of **1** and **2** with the coordination environment of the Zn(II) atom was confirmed by single-crystal X-ray diffraction (Figure 4). In detail, for **1**, the Zn(II) ion is set on inversion centers that were chelated by one triazole-carboxylate ligand. Two nitrogen symmetry-related atoms of the triazole and pyridyl fragments build the equatorial plane around the Zn(II) ions for **1** (Zn1−N4 2.151(3) Å, Zn1−N1 2.297 (3) Å). Meanwhile, the central Zn(II) ion also was coordinated by two oxygen symmetry-related atoms from carboxylate fragments at the equatorial axial positions (Zn1−O1 2.044(2) Å), giving rise to its octahedral geometry, with the *trans*-N_4_O_2_ configuration. For compound **2**, the Zn (II) ion is located on a two-fold-axis, that was chelated by one triazole-carboxylate ligand. Two nitrogen symmetry-related atoms of the triazole and pyridyl fragment build the equatorial plane around the Zn(II) ions (Zn1−N4 2.166(3) Å, Zn1−N1 2.142 (3) Å). Meanwhile, the central Zn(II) ion also was coordinated by two oxygen symmetry-related atoms from carboxylate fragments at the axial positions (Zn1−O1 2.151(2) Å), giving rise to its octahedral geometry, with the *cis*-N_4_O_2_ configuration. The N–Zn–N and O–Zn–O bond angles are in the ranges 83.00(9)-97.00(9)/88.70(16)-163.94(12)°;180/174.64(12)° for **1** and **2**, respectively. The bond distances and angles of **1** and **2** (Appendix A) are similar to those reported in the literature for related compounds, with N_4_O_2_ donors set as representative examples. In each case, the asymmetric unit comprises half of a complex molecule.

In **1** and **2**, the structural analysis also revealed the presence of [2+2] polymeric metallocycle complexes with two ligand molecules coordinated to a pair of symmetry-related Zn(II) ions, resulting in the formation of 16-membered metallo-cyclic rings, and then forming a rhomboid fashion, with a cross-sectional area of 30.68 and 38.77 Å^2^ (Figure 5). The so-formed metallocycles show different coordination modes due to the isomeric effect of the ligand over the metal center (*cis and trans* for **1** and **2**, respectively). This effect is also reflected in the orientation of the triazole and *n-*pyridyl rings, concerning the rhomboid shape, meaning the plane generated by Zn1 and C6 atoms and their symmetry-related homologs atoms in each compound. The dihedral angles between the metallacycle mean plane and the triazole ring are 65.34(10) and 84.79(12)°, respectively. Likewise, the dihedral angles between the metallacycle mean plane and the *n-*pyridyl rings are 68.73(18) and 54.25(14)°, respectively. The Zn⋅⋅⋅Zn separations in the mean rhomboid plane are 9.004(5)Å and 9.315(3) for **1** and **2**, respectively. Therefore, **1** and **2** form a 1D coordination polymer chain along the [101] and [010] directions, respectively (Figure 6). Each chain is stabilized by an intramolecular hydrogen bond involving C atoms from the methyl group and O atoms of the carboxylate group, *viz.* C10–H10B⋅⋅⋅O1 (H10B···O1 2.60 /2.62 Å, C10–O1 3.241(4)/ 3.241(5) Å, C10–H10B⋅⋅⋅O1 124.5/122.9°) for **1** and **2,** respectively. For **1**, these interactions form a centrosymmetric ring. For **2**, other intramolecular hydrogen bond interactions are observed between C atoms of the pyridyl ring and one water molecule, *viz*. C4–H4⋅⋅⋅O2S (H2SA···O1S 2.17Å, O2S–O1S 3.020(6)Å, O2S−H2SA···O1S 173.7°) (Figure 5).

The supramolecular structure reveals that **1** and **2** have 2D networks that are induced by hydrogen bonds. In **1**, the 2D supramolecular aggregate is built by single weak hydrogen bond interactions between oxygen and hydrogen atoms from organic molecule neighbors, *viz.* C4–H4⋅⋅⋅O1 [H4⋅⋅⋅⋅O1 2.48 Å, C4-H4-O1 130.0°] (Appendix A) to form chains of C(8) running along [010]; moreover this kind of interaction forms a centrosymmetric metallocycle with a graph-set motif, (16) (Figure 5). In compound [Zn(L2)]⋅4H_2_O (**2**), the water molecules link a 1D polymeric chain forming a 2D supramolecular aggregate. A 1D polymeric chain is linked to the neighboring chain by O1S–H1SB⋅⋅⋅O1 and O1S–H1SB⋅⋅⋅O2, generating (4) non-centrosymmetric rings (labeled A); O1S–H1SB⋅⋅⋅O1 and O1S–H1SA⋅⋅⋅O1, generating (8) centrosymmetric rings (labeled B); O1S–H1SB⋅⋅⋅O1; and O1S–H1SB⋅⋅⋅O2 and O2S–H2SB⋅⋅⋅O1S, generating (19) centrosymmetric rings (labeled C) (Figure 6). The water molecules are linked to form a tetrameric aggregate by O2S–H2SB⋅⋅⋅O1S and O2S–H2SA⋅⋅⋅O1S hydrogen bond interactions, generating a (8) centrosymmetric ring (labeled D) (Figure 5). These four types of rings alternate in an ABACDC fashion to form a two-dimensional supramolecular aggregate (Figure 6 and Appendix A).

### 2.3. Luminescent Properties

The room temperature solid-state emission spectra, emission data, and electronic absorption of Na**L1**, Na**L2**, **1**, and **2** are shown in Figure 7 and Table 1. These crystalline solids have interesting luminescent properties, with significant differences in their spectra.

Thus, compounds Na**L1** and Na**L2** show significant differences in their emission intensity and a large redshift, Δλ = 41 nm, between Na**L1** and Na**L2**. These differences are attributable to their isomerism and the bonding mode with the Na(I) counterion. It is well-known that the ionic strength in the luminescent ionic compounds can quench the emission response [51]. Therefore, it can be assumed that the difference in the intensity and the shift in the emission band between Na**L1** and Na**L2** is due to the interaction between the sodium counterion and the ligand. On the other hand, as the emission bands for the Na**L1** and Na**L2** are around 420–450 nm, the involved transitions are mainly π–π*. Moreover, at about 630 nm, a small band is observed in Na**L2**, which can be attributable to counterion-ligand charge transfer (XLCT), supporting the hypothesis of the decrease of the intensity emission maximum between Na**L1** and Na**L2** [31].

In **1** and **2**, a different situation is observed compared with their respective ligands. For instance, the metal coordination clearly produces red shifting and intensity changes. These changes are due to the chelation of the metal center that generates an increase in the rigidity of the respective coordination polymers. These structural constraints avoid energy loss by the nonradiative decay of the intra-ligands excited states [52]. The structures **1** and **2** show emission bands centered at 367 and 366 nm, respectively (Figure 7). However, **2** has a more significant luminescent response (~1000 times greater). As explained above, the difference in intensities is due to the rigidity that the metal ion contributes to the formation of these systems. The presence of electronically saturated *d*^10^ ions, such as Zn(II), are appropriate, because they impose conformational rigidity to the ligand prevent energy loss via bond vibration or electron transfer processes [53,54]. Although both ligands are relatively similar, the isomeric positional effect can generate the difference in the coordination modes over the metal center (*cis versus trans*) that may contribute to a loss of intensity. The centrosymmetric setting **1** may be less favorable in the luminescent response than **2**, due to the Zn(II) ion bond distances being smaller in **1** than in **2** (Appendix A), generating a possible quenching by the concentration in the solid state of the different distances between Zn (II) ions that could effect this [55,56].

### 2.4. Thermogravimetry (TG) Analyses

The thermogravimetric analyses of Na**L1**, Na**L2**, **1**, and **2** were recorded with a heating rate of β = 10 °C·min^−1^ under a dynamic nitrogen atmosphere at 20–700 °C. All curves are shifted to a higher temperature at a constant heating rate.

Na**L1** and Na**L2** are stable up to ca. 322 and 306 °C, respectively. After these temperatures, the progressive decomposition of the ligands occurred. The organic fragments completely decompose at 462 °C with a loss of 49% weight for Na**L1**, and 630 °C with a loss of 57% for Na**L2**. The black color residue remains at the end of the heating process and contains Na_2_O and carbonaceous matter (Figure 8).

In the case of **1**, the TG curves show a four-step weight loss until total decomposition, starting at ca. 30 °C, with a total deterioration of over 500 °C. The first step corresponds to the loss of two water molecules due to moisture in the sample (~6%). In the second step at 300 °C, the decarboxylation from the ligand was found (~8%). The two following steps correspond to the progressive decomposition of the compound. The TG of **2** shows a two-step decomposition curve. The first one at ~110 °C represents a weight loss of 5% (two water molecules). The second step at ca. 290 °C corresponds to the progressive decomposition of the organic ligand. After the total disintegration of both CPs, the final compounds correspond to ZnO and carbonaceous material (Figure 8 and see Appendix A, for more details).

## 3. Materials and Methods

### 3.1. Materials and General Procedures

All reagents used were purchased from Sigma-Aldrich (Sigma-Aldrich, St. Louis, MO, USA) and used without further purification. 

### 3.2. Characterization

FT-IR spectra in the range 400–4000 cm^−1^ were recorded on a Nicolet Avatar 300 spectrometer (Thermo Scientific,Waltman, MA, USA) using KBr pellets. NMR spectra were recorded at 298 K with a Bruker Avance III-HD Nanobay 300 MHz (Bruker Co.; Billerica, MA, USA). All NMR spectra are reported in parts per million (ppm, d) relative to tetramethylsilane (Me4Si) for 1H and 13C NMR spectra, with the residual solvent proton and carbon resonances used as internal standards. Coupling constants (J) are reported in hertz (Hz), and integrations are reported as the number of protons. The following abbreviations are used to describe peak patterns: s = singlet, d = doublet, t = triplet, m = multiplet, br = broad. High-resolution electrospray ionization mass spectra (ESI-MS) were obtained on a Thermo Fisher scientific ultimate 3000 y Q exactive focus mass spectrometer (Thermo Scientific, Waltman, MA, USA) and the results are expressed in a mass/charge ratio (m/z) in a positive mode. The thermogravimetry (TG) analyses were carried out on a STA 448 Jupiter F3 type simultaneous thermal analyzer (Netzsch, MA, USA). For TG, 6 mg of the samples were used as microcrystalline powders. The used sample cells were aluminum oxides pans. The parent reagents were heated up to 700 °C at a heating rate of 10 °C min^−1^ under a flow of nitrogen at 20 mL min^−1^. Excitation (PLE) and emission (PL) spectra were measured at room temperature using a Jasco FP-8500 spectrofluorometer with a 150 W xenon lamp as the excitation. The luminescence spectra were recorded on a JASCO FP-8500 spectrofluorometer JASCO Co.; Kyoto, Japan) in the solid-state at room temperature. The excitation was performed with λex = 250–400 nm, and the emission was recorded at λem = 400–750 nm. All spectra were measured with 0.05 mmol of each compound.

### 3.3. X-ray Powder Diffraction

The X-ray powder diffraction diffractograms of complexes **1** and **2** were collected at room temperature, in a Panalytical X’Pert Pro automated diffractometer (Malvern Panalytical, Moreira, Portugal) equipped with an X’celerator detector by using CuKα radiation (λ = 1.54177 Å). The diffractometer was operated at 40 kV and 40 mA in θ/θ reflection mode. The powder patterns were scanned in the range of 2θ = 5-50°, with a step size of 0.02° and a counting time of 20 s per step (Appendix A).

### 3.4. Single-Crystal X-ray Diffraction

Some suitable single crystals of each compound were measured. Their diffraction data were collected at 293-295 K on a Bruker D8 Venture diffractometer equipped with a bidimensional CMOS Photon 100 detector, using graphite monochromated Cu-Kα (*λ* = 1.54178 Å) radiation. The diffraction frames were integrated using the APEX3 package [57] and were corrected for absorptions with SADABS. The structures of all compounds were solved by intrinsic phasing [58] using the OLEX 2 program [59]. The structures were then refined with full-matrix least-squares methods based on *F*^2^ (*SHELXL-2014*) [58]. For the four compounds, non-hydrogen atoms were refined with anisotropic displacement parameters. All hydrogen atoms were included in their calculated positions, assigned fixed isotropic thermal parameters, and constrained to ride on their parent atoms. A summary of the details about crystal data, collection parameters, and refinement are documented in Table 2, and additional crystallographic details are provided in the CIF files. ORTEP views were drawn using OLEX2 software [59].

### 3.5. General Procedure for the Syntheses of NaL1 and NaL2

The precursor of the organic ligands was prepared according to standard methods reported in the literature [28,42], generating the precursor ester compounds ethyl 5-methyl-1-(pyridin-3-ylmethyl)-1H-1,2,3-triazole-3-carboxylate (**P1**) and ethyl 5-methyl-1-(pyridin-3-ylmethyl)-1H-1,2,3-triazole-4-carboxylate (**P2**). The esters were saponified with a solution of NaOH (2 eq.) in a minimum amount of methanol to generate the corresponding carboxylate sodium salts-compounds Na**L1** and Na**L2**, respectively (Figure 1).

Na**L1**: Yield: quant.98% IR (KBr, cm^−1^); ν: 3037(m), 3006(m) (Csp^2^-H); 2968(m), 2952(m) (Csp^3^-H); 1619(s), 1598(vs) (C=O-), 1565(s) (N=N); 1486(s), 1457(s), 1432(vs) (Csp^2^-Csp^2^); 1415(s) (COO-as). HR-ESI-MS for C_9_H_7_N_4_O_2_Na [NaL^−^]^+^: calculated = 217.073, found = 217.0727 (1 ppm). ^1^H NMR (300 MHz, DMSO) δ 8.52 (dd, J = 4.8, 1.6 Hz, 1H, H-A), 8.48 (d, J = 1.7 Hz, 1H, H-B), 7.54 (dt, J = 3.9, 1.8 Hz, 1H, H-D), 7.39 (ddd, J = 7.9, 4.8, 0.8 Hz, 1H H-C), 5.55 (s, 2H; =C-CH_2_-N=), 2.43 (s, 3H; -CH_3_).

Na**L2**: yield: quant.98% IR (KBr, cm^−1^); ν: 3070(m), 3049(m) (CAr-H); 2985(m), 2927(m) (Csp^3^-H); 1606(vs) (C=O^−^), 1562 (s) (N=N); 1500(m), 1475(m)(Csp^2^-Csp^2^); 1419 (s) (COO-as). HR-ESI-MS for C_9_H_7_N_4_O_2_Na [NaL^−^]^+^: calculated = 217.0731, found = 217.0721 (1 ppm). ^1^H NMR (300 MHz, DMSO) δ 8.55 (dd, J = 4.4, 1.6 Hz, 1H; H-A), 7.06 (d, J = 6.0 Hz, 1H; H-B), 5.58 (s, 2H; =C-CH_2_-N=), 2.39 (s, 3H; -CH_3_).

### 3.6. General Procedure Syntheses for 1D CPs [Zn(L1)]n (**1**) and [Zn(L2)·4H_2_O]n (**2**)

A solution of Zn(NO_3_)_2_·4H_2_O (34 mg, 0.115 mmol) in *n*-butanol (5 mL) was slowly added over an aqueous solution of Na**L** (Na**L1** or Na**L2**) (5 mL, 5 mg, 0.230 mmol). The resulting mixture was stored at room temperature for two weeks. Then, the formation of crystals suitable for SC-XRD analysis from the interphase occurred. The crystals were collected by hand and air dried. **1**: Yield: 30%. IR (KBr, cm^−1^); ν: 3012(w) (C*sp*^2^-H); 2977(w) (C*sp^3^*-H); 1633(s) (C=O^−^), 1610(vs) (C=N^−^), 1581(s) (N=N); 1484(m), 1463(m), 1430(m) (C*sp*^2^-C*sp*^2^); 1384(s) (COO^−^as). **2:** Yield: 40%. IR (KBr, cm^−1^); ν: 3070(w), 3016(w) (C*sp*^2^-H); 2973(w), 2929(vw),2923(vw) (C*sp^3^*-H); 1664(s) (C=O-), 1618(vs) (C=N-), 1567(s) (N=N); 1484(m), 1427(m) (C*sp*^2^-C*sp*^2^); 1384(vs) (COO^−^as). The X-ray powder diffractions of **1** and **2** confirm that the powders and single crystals have the same structural phase (Appendix A).

## 4. Conclusions

In summary, two new CPs with the semi-rigid *n*-methylpyridyltriazole carboxylate ligands, with *n* = 3 (**L1**) and 4 (**L2**), have been prepared at the water/*n*-butanol interphase by reacting Zn(NO_3_)_2_·4H_2_O with Na**L1** and Na**L2**. Their X-ray structures confirm the formation of two 1D coordination polymers in which the flexibility of the methyl group and the change in the N-donor portion of the pyridyl entity *n*-methylpyridyltriazole carboxylate ligands coordinate to the metal center in a *cis versus trans* conformation, giving rise to the formation of rather different H-bond supramolecular networks. Remarkably, **2** shows a strong emission centered at 367 nm at room temperature. The structural differences result in the centrosymmetric setting **1** being less favorable in the luminescent response than **2**, due to the different distances between Zn (II) ions that could cause a quenching by concentration in the solid state.

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
