# Peer review of "The Methylene Spacer Matters: The Structural and Luminescent Effects of Positional Isomerism of n-Methylpyridyltriazole Carboxylate Semi-Rigid Ligands in the Structure of Zn(II) Based Coordination Polymers"

_polymers, 2023, doi:10.3390/polym15040888_

Round 1
Reviewer 1 Report
The manuscript by Brito and co-workers highlighted two Zn(II) coordination polymers based on semi-rigid organic ligands. Furthemore, the authors mentioned pivotal role of non-covalent interactions to afford 2D network and characterized using Single-crystal X-ray diffraction. This work is of interest to reader and could be suitable for Polymers, MDPI after consideration of the following points:
1. The authors should highlight the rationale of choosing methylene spacer linkers and explain the affect of increasing flexibility i.e. from methylene to ethylene.
2. The Rwp and Rp values of refinement should be included.
Author Response
- Question The authors should highlight the rationale of choosing methylene spacer linkers and explain the affect of increasing flexibility i.e. from methylene to ethylene.
Answer The Incoporation of the methylene group on our ligands allows to obtain semi-rigid ligands in CPs generating the formation of more diverse structures compared to rigid linking ligands. We are currently working with the incorporation of spacers of the type (-CH2-)n with n=2-5, which will allow us to study with more details, the effect of the flexibility of the spacer on the physicochemical properties of the coordination polymers or MOFs obtained.
- Question: The Rwp and Rp values of refinement should be included.
Answer: The Rwp and Rp values of refinement, were included the supplementary material (see PDRX patterns) Rp= 6.6, Rwp= 8.7 (for compound 1) Rp= 5.9, Rwp= 7.8 (for compounds 1 &2 respectively)
Reviewer 2 Report
Dear Editor,
Brito et al. in this work describe the preparation of two CPs based on Zinc and two semi rigid organic linkers. They provide details regarding both the synthesis of the organic linkers and the CPs. Regarding the structural characterization of the CPs they provide a detailed crystallographic characterization explaining the importance of the -CH2- spacer regarding the structure conformation. Also, they discuss the importance in the properties of the material. The phase purity of the synthesized CPs is proved through PXRD diagrams compared with the calculated patterns from the SC-XRD data. Furthermore, the physicochemical properties of the materials are discussed regarding their thermal stability through TGA experiments as well as their luminescent properties which are described in detail. In general, the experimental part of the manuscript is clear and the experimental procedures along with the obtained results are detailed and sound. On the other hand, as far as I am concerned the introduction and the references could be improved, however I do believe that after a few revisions this work would be suitable for publication in Polymers.
In detail my comments are the following:
1. Line 24, the molecular formula should be revised regarding the proper presentation of subscripts.
2. In the first paragraph of the introduction the authors attempt to describe the building units of a CPs using the terms spacers and nodes. These terms are widely used in MOF terminology. Although MOFs can be included in the general family of CPs it is not clear whether the authors would like to describe their compounds as MOFs or not. Therefore, I would suggest adding a small paragraph making this part clearer to the reader.
3. Also, I would suggest that the authors provide a few more information regarding the importance of Zinc metal centers to their compounds. For example, other literature examples from other scientific groups especially regarding their effect in the luminescent properties.
4. Although the crystallographic description of the compounds is very detailed and provides the reader with all the appropriate information. I would suggest that they would also provide the check CIF files to make their results more sound.
5. In the text the authors claim that these CPs are stable in moisture. Are there any experimental data for their claim. For example, a PXRD of the CPs after their exposure in air for prolonged time.
6. In line 232, I would suggest the addition of literature references regarding the rigidity that the metal ion provides.
7. In the description of the TGA the authors claim that ZnO is formed. Are there any experimental data for that, perhaps a PXRD?
8. Also, I would like to comment regarding the number of self-references in some paragraphs, I believe it would contribute to the image of their work to accompany these parts also with references from diverse scientific groups.
Author Response
-
- Question1: Line 24, the molecular formula should be revised regarding the proper presentation of subscripts.
Answer: In the final manuscript the molecular formula was changed to Zn (NO3)2·4H2O
- Question 2 In the first paragraph of the introduction the authors attempt to describe the building units of a CPs using the terms spacers and nodes. These terms are widely used in MOF terminology. Although MOFs can be included in the general family of CPs it is not clear whether the authors would like to describe their compounds as MOFs or not. Therefore, I would suggest adding a small paragraph making this part clearer to the reader.
Answer: In the final manuscript, we have included the next text: Considering that our metal-organic compounds have 2D dimension we will refer to them as coordination polymers. (line- 48-49)
- Question 3 Also, I would suggest that the authors provide a few more information regarding the importance of Zinc metal centers to their compounds. For example, other literature examples from other scientific groups especially regarding their effect in the luminescent properties.
Answer : we have included the the next text: Additionally a great attention is focused on the study of luminescent Zn(II) complexes in the solid state and in solutions. Luminescent zinc(II) complexes are used for production of organic light-emitting diodes (OLED) [32-37]. (lines 65-67)
- Question: Although the crystallographic description of the compounds is very detailed and provides the reader with all the appropriate information. I would suggest that they would also provide the check CIF files to make their results more sound.
Answer: in the supplementary material section, there are the csd deposit numbers of all the compounds, where the reader can consult the chekcif of each compound described in the manuscript, among other data.
5 Question In the text the authors claim that these CPs are stable in moisture. Are there any experimental data for their claim. For example, a PXRD of the CPs after their exposure in air for prolonged time.
Answer: We state that indeed XRPD diffraction patterns were measured at the CPs months after their preparation and they showed the same x-ray powder original patters
6 Question In line 232, I would suggest the addition of literature references regarding the rigidity that the metal ion provides.
Answer; We have included in final manuscript the next text: The presence of electronically saturated d10 ions, as Zn(II), are appropriate, because they impose conformational ridigity to the ligand and prevent energy loss via bond vibration or electron transfer processes [49,50]. (lines 248-250)
7 Question: In the description of the TGA the authors claim that ZnO is formed. Are there any experimental data for that, perhaps a PXRD?
Answer: We have made a PDRX after total decomposition of compounds 1 & 2 both compounds shown the typical diffraction patters for ZnO (see figure S2 in the supplementary material). We have included a phrase indicating the figure in the supplementary material.
8. Question Also, I would like to comment regarding the number of self-references in some paragraphs, I believe it would contribute to the image of their work to accompany these parts also with references from diverse scientific groups.
Answer: In the final manuscript we have included information relative to our previous works, including more relatated references of other authors (lines 68-71).
Round 2
Reviewer 2 Report
Dear Editor,
The authors have addressed successfully the majority of the comments. However, I must mention that the issue with the CCDC numbers that they provide do not match with a CIF file in the CCDC database. More specifically I refer to numbers 2175399, 2175400, 2175401 and 2175402. I am not sure if there is a problem with the CCDC platform or a typo mistake. That is why I initially asked for the check cif file, because in the present situation I can not express a valid opinion regarding the crystallographic data. Unfortunately I can not approve the article for publication until this issue is resolved.
Author Response
Dear Reviewer,
Thanks you very much for your suggestion.
Concerning your comments, we state the following:
CCDC deposition number are correct, but you can not check the CIFs files, because this are not available yet. This is due to the CIF file is released when the manuscript be published or those be embargoed. In both situation CIF files will be available and become public domain.
You can check the attached document as proof the we are saying, this also include the checkCIF of each compound.
We also include a little sentence in the supplementary material section in the manuscript document to avoid misunderstanding in the reading.
Best regards

Round 3
Reviewer 2 Report
The authors have answered all of the comments. This work as far as I am concerned is suitable for publication in polymers
Author Response
Regarding the files, please find in the supplementary materials that the CIF files of the structures that were deposited, downloaded directly from the CCDC, Thanks in advance for your comments and suggestions.